

# Gaze behavior and decision-making among handball referees: exploring gender and expertise differences

Jacek Świdwa[1], Stefanie Klatt[2] and Adam Kantanista[1]

[1] Department of Physical Education and Lifelong Sports, Poznan University of Physical Education, Poznan, Poland
[2] Institute of Exercise Training and Sport Informatics, Section Cognition in Team Sports, German Sport University Cologne, Cologne, Germany

## ABSTRACT

**Background**. Gaze behavior has been extensively studied in various sports, yet research on handball referees remains limited. Understanding gaze behavior in handball officiating is crucial for enhancing training programs, particularly for novice referees. This study investigates gaze behavior and decision-making processes among male and female handball referees of varying expertise levels.

**Methods**. A total of 51 handball referees (aged $30.25 \pm 7.61$ years), including 11 females and 40 males from the Polish Handball Federation, participated in the study. The sample comprised 31 higher-level referees (Super League and First League) and 20 lower-level referees (Second and Regional League). Participants wore head-mounted mobile eye-trackers to assess fixations and saccades while watching video clips of handball match scenarios. After each scene, referees made decisions based on the handball rules.

**Results**. Higher-level referees demonstrated significantly greater decision-making accuracy compared to lower-level referees ($p < 0.05$; Cohen's $d = 0.678$), particularly in "punishment" scenarios ($p < 0.001$; Cohen's $d = 1.407$). Although no significant differences in gaze behavior (*e.g.*, number and duration of fixations and saccades) were observed concerning gender or expertise level, specific differences in decision-making accuracy emerged, particularly regarding expertise and free-throw scenarios.

**Conclusion**. The findings indicate that differences in decision-making accuracy among handball referees are likely influenced by factors such as experience and cognitive processing rather than gaze behavior. The absence of gender differences in gaze patterns challenges prior research suggesting systematic visual search disparities. Future studies in real-game settings are needed to explore the impact of physical and psychological demands on referees' performance, providing practical insights for training programs.

Corresponding author
Jacek Świdwa, swidwa@awf.poznan.pl

# INTRODUCTION

Handball, an Olympic team sport, involves dynamic offensive and defensive actions such as running, jumping, throwing, and blocking (*Gorostiaga et al., 2006*) and places significant demands on referees. These officials play a crucial role in ensuring fair play and adherence to the rules of the sport (*Dohmen & Sauermann, 2016*), requiring them to make swift and

accurate decisions amidst the fast-paced nature of the game (*Mascarenhas & Smith, 2011*). This responsibility necessitates not only sharp visual and auditory acuity but also physical fitness, emotional composure, and high levels of motivation (*Bichot, Rossi & Desimone, 2005*; *Belcic, Ruzic & Marosevic, 2020*; *Belcic, 2022*; *van Biemen et al., 2022*).

Refereeing entails decision-making under time constraints, demanding a range of skills including rapid reaction times, anticipation, concentration, and effective collaboration with fellow officials (*Pietraszewski et al., 2014*). Numerous factors influencing referee decision-making have been explored, including physical activity (*da Silva et al., 2010*; *Ahmed, Davison & Dixon, 2017*; *Belcic, Ruzic & Marosevic, 2020*), psychological demands (*Diotaiuti et al., 2017*; *Dodt, Fasold & Memmert, 2022*), body composition (*Belcic, 2022*), social dynamics, body height (*McCarrick et al., 2020*), and environmental factors (*Johansen & Erikstad, 2020*). While most of the research has concentrated on football (*Lane et al., 2006*), certain findings may be applicable to other team sports (*e.g.*, handball) where refereeing decisions can also be made by referees performing different functions, and the game has a contact nature.

The significance of expertise levels in both athletes and referees underscores the complexity of decision-making processes in team sports (*Hüttermann, Memmert & Simons, 2014*). Recent studies have demonstrated that expertise influences referees' decision accuracy by enabling them to anticipate game situations more effectively and allocate attentional resources more efficiently (*Schrödter et al., 2024*). Advancements in technology, particularly the use of eye-tracking devices, have greatly enhanced our understanding of these processes. Eye-tracking technology has been utilized to study learning process (*Cates & Gordon, 2022*), refine research methodologies (*Meyer et al., 2022*), and analyze participants' perceptions in real-world game scenarios (*Fasold et al., 2018*). Unlike previous studies, the present research specifically investigated the quantity and temporal characteristics of handball referees' fixations and saccades in a controlled laboratory setting. Gaze behavior has been recognized as a valuable tool for tracing decision-making processes (*Hancock & Ste-Marie, 2013*).

Expertise is often associated with more efficient visual processing, as described by the information-reduction hypothesis (*Haider & Frensch, 1999*), which posits that skilled individuals focus selectively on task-relevant information while ignoring redundant cues. This has been supported in sports refereeing by findings that higher-level referees demonstrate superior cognitive strategies, such as faster anticipation and better decision-making accuracy (*Catteeuw et al., 2009*; *Hancock & Ste-Marie, 2013*). For example, *Hüttermann et al. (2018)* found that football assistant referees with greater attentional capacity made significantly fewer offside errors, highlighting the crucial role of visual attention in officiating decisions.

However, the literature on gaze behavior differences between referees of varying expertise is mixed, with some studies reporting significant differences (*e.g.*, *Schnyder et al., 2014*) and others finding no differences (*Hancock & Ste-Marie, 2013*). Recent work has also suggested that referees' perspective and viewing angles influence their decision accuracy, with *Schrödter et al. (2024)* demonstrating that first-person perspectives lead to higher certainty and improved recognition of no-calls compared to third-person perspectives in

video-based training. Based on the theoretical framework of expertise and prior evidence indicating potential advantages in decision-making accuracy at higher skill levels, we hypothesized that higher-level referees would demonstrate longer fixation durations, fewer saccades, and superior decision-making accuracy compared to their lower-level counterparts. This hypothesis aligns with the idea that expertise enables referees to allocate visual attention more efficiently and interpret game scenarios more effectively.

The present study builds on prior research by examining not only decision-making accuracy but also the temporal and quantitative aspects of gaze behavior among handball referees. Unlike prior studies that predominantly focus on football or ice hockey referees, this research investigates these dynamics in handball, a fast-paced team sport with unique visual and cognitive demands. By combining these perspectives, we aim to contribute to a more comprehensive understanding of how expertise shapes visual attention and decision-making in refereeing, particularly in underrepresented sports like handball.

To the best of our knowledge, this is the first study to explore decision-making and gaze behavior among handball referees in a controlled laboratory setting. We identified only one previous study (*Fasold et al., 2018*) that analyzed the coordinated gaze behavior of a team of professional referees during a live handball match. However, the objectives and methods of that study differed significantly from the present research. *Fasold et al. (2018)* focused on identifying where field and goal referees directed their gaze during critical attacking phases of the match, without examining performance outcomes or decision-making accuracy. Their findings revealed that the referee team predominantly fixated on the same elements of gameplay or the same areas of the court during these phases. This suggests that the referees did not consistently adhere to practice-oriented guidelines, which emphasize distributing attention across various aspects of the game rather than concentrating on the same elements.

Additionally, stress and game management strategies have been identified as crucial factors in referee decision-making. *Rückel et al. (2021)* demonstrated that elite volleyball referees employ various coping strategies to maintain performance under stress, suggesting that psychological resilience plays a key role in officiating performance. These findings align with those of *Klatt et al. (2025)*, who highlighted that increased physiological stress responses can negatively impact cognitive control in referees. Given that referees operate under high-pressure environments, it is essential to explore whether stress responses impact their gaze behavior and decision accuracy.

Research has investigated the gaze behavior patterns of referees across various sports, with a particular focus on their decision-making abilities. In ice hockey refereeing, *Hancock & Ste-Marie (2013)* found no differences in gaze behavior between lower- and higher-level referees. However, more experienced officials demonstrated greater accuracy and sensitivity in their decisions-making processes. Similarly, studies by *Catteeuw et al. (2009)* and *Catteeuw et al. (2010)* investigated decision-making among assistant referees in football, revealing that differences primarily emerged in decision accuracy. While assistant referees collectively spent time fixating on the offside line, more experiences assistant referees directed their gaze toward the receiving attacker. Although these differences did not reach

statistical significance, they suggest subtle variations in the visual strategies employed by officials at different expertise levels.

Another study examining offside situations showed that visual search behavior varied based on distance and visual angle in these game scenarios (*Luis del Campo et al., 2018*). The authors reported that referees performed a higher number of fixations and spent more time fixating on the player with the ball when observing the game from larger visual angles compared to smaller ones. In addition, referees fixated more frequently and for longer durations on the last defender while reducing both the number and duration of fixations on the player receiving the ball when the action occurred closer to the referee rather than at greater distances.

*Schnyder et al. (2014)* reported similar findings in football refereeing, highlighting that expert-level officials demonstrated superior decision-making compared to their near-expert counterparts. Interestingly, no significant differences were observed in gaze behavior, as referees exhibited similar fixation locations. Further research found that elite football referees employed a lower search rate and were more likely to focus their gaze on the ball during the moment of the kick and the early flight phase of the pass. Additionally, a significant relationship was identified between fixation duration and referees' level of experience. It is suggested that more experienced referees subsequently produced earlier anticipatory eye movements toward the players receiving the ball (*van Biemen et al., 2022*).

Studies examining the same sport discipline yield different results. *Spitz et al. (2016)* found discrepancies in fixation locations between elite and sub-elite referees during open play situations and corner kicks. Experienced referees exhibited longer fixation durations in the contact zone during these scenarios compared to their sub-elite counterparts. Interestingly, both groups fixated on similar locations; however, decision accuracy differed. Similarly, *Klatt et al. (2021)* examined referees' gaze behavior in basketball and found that officiating teams allocate visual attention efficiently, aligning with recommendations for optimal decision-making strategies.

Research comparing decision-making between male and female referees, particularly in the context of gaze behavior across various sports, remains limited. This scarcity is likely due to the challenge of obtaining a representative sample of female referees, a consequence of their underrepresentation in the field. In many studies involving male and female handball referees, participants were analyzed collectively, without gender-specific comparisons. Some researchers, such as *Diotaiuti et al. (2017)*, have made efforts to include both male and female referees in their samples, aiming for a more comprehensive understanding of the challenges referees face regardless of gender.

Beyond decision-making contexts, studies on gender-based differences in gaze behavior reveal notable distinctions. Empirical evidence suggests that women exhibit more exploratory visual patterns, characterized by larger saccade amplitudes and extended scan paths, facilitating faster image inspection compared to men. Research also highlighted the importance of regions of interest (ROIs), relevant to experimental tasks involving referees. For example, during the observation of human faces, gender differences in eye movement patterns were not attributed to variations in fixation frequency but rather to distinct saccadic trajectories. Specifically, women showed a significant increase in transitions from

other facial areas to the eyes compared to men. Given these findings on gender-related differences in gaze behavior (*Sammaknejad et al., 2017*; *Abdi Sargezeh, Tavakoli & Daliri, 2019*), the present study sought to examine this factor in the context of handball refereeing.

Previous studies investigating the relationship between gaze behavior and referees across various sports have typically been limited by small sample sizes, ranging from 6 to 39 participants. Moreover, these studies either exclusively focused on male referees or did not provide information about the referees' gender (*Ziv et al., 2020*). In our study, we aimed to address this gap by examining differences in gaze behavior and decision-making among handball referees, stratified by expertise level and gender. Although our study includes a larger sample size compared to previous research in this area, the primary focus is on addressing the lack of studies examining both gaze behavior and decision-making among handball referees. This dual focus highlights a significant gap in the literature, particularly in the context of gender and expertise differences in officiating.

We hypothesized that there would be no discernible differences in gaze behavior and decision-making accuracy between female and male handball referees. This hypothesis is based on previous findings suggesting no consistent gender-based differences in overall gaze behavior or decision-making accuracy. However, subtle differences in visual search strategies and cognitive processing have been reported in some studies (*e.g.*, *Abdi Sargezeh, Tavakoli & Daliri, 2019*; *Sammaknejad et al., 2017*). Investigating this claim addresses a significant gap in the literature, particularly given the underrepresentation of female referees in sports officiating research.

Additionally, we hypothesized that referees at higher levels would exhibit a greater frequency of accurate decisions in relation to game situations. This expectation is supported by the information-reduction hypothesis (*Haider & Frensch, 1999*), which posits that experienced individuals are better at focusing on task-relevant information while ignoring redundant cues. Prior research also suggests that higher-level referees possess superior cognitive strategies, enabling them to interpret visual information more effectively (*Catteeuw et al., 2009*; *Hancock & Ste-Marie, 2013*).

## MATERIALS & METHODS

### Participants

The study encompassed 51 handball referees (mean age 30.25 ± 7.61 years) affiliated with the Polish Handball Federation, representing approximately 20% of all referees within the federation. This sample comprised 40 men (mean age 30.55 ± 8.10 years) and 11 women (mean age 29.18 ± 5.70 years). Referees officiating in the Super League and First League were categorized as the higher-level referee group (men: $n = 24$, mean age = 33.83 ± 6.89 years; women: $n = 7$, mean age = 32.57 ± 2.64 years), while those in the Second and Regional League were classified as lower-level referee group (men: $n = 16$, mean age = 25.63 ± 7.39 years; women: $n = 4$, mean age = 23.25 ± 4.57 years). Participants had an average refereeing experience of 10.52 ± 5.80 years (men = 11.12 ± 5.98 years, women = 8.36 ± 4.69 years). All participants were active referees registered with the Polish Handball Federation and volunteered for the study, providing informed consent.

**Table 1  Basic characteristics of participants.**

| | Women (n = 11) | Men (n = 40) | p-value | Higher-level referees (n = 20) | Lower-level referees (n = 31) | P value |
|---|---|---|---|---|---|---|
| | M ± SD | | | M ± SD | | |
| Experience (years) | 8.36 ± 4.69 | 11.12 ± 5.98 | 0.290 | 13.26 ± 5.22 | 6.30 ± 3.81 | <0.001 |
| Age (years) | 29.18 ± 5.71 | 30.55 ± 8.10 | 0.774 | 33.55 ± 6.17 | 25.15 ± 6.88 | <0.001 |

Notes.
M, mean; SD, standard deviation.

Written consent from all participants was obtained. The study involved only participants without visual impairments. Basic characteristics of participants are presented in Table 1. The project was approved by the Ethics Committee at the Poznan University of Medical Sciences (KB/357/21) and adhered to the ethical standards outlined in the Declaration of Helsinki.

## Procedure and test protocol

At the beginning, all participants completed a personal questionnaire to provide details about refereeing experience and handball background. During data collection, only the participant and evaluator were present in the testing room. Participants viewed handball scenes projected on a 60-inch overhead screen.

Prior to the test, the researcher provided a comprehensive explanation of the procedure, which involved observing handball scenes depicting actions occurring within 6 and 9 m from the perspective of a goal line referee. Participants were informed that they would make decisions after each scene, and were told the test would last approximately six minutes. They were also instructed to maintain a stable head and neck position to ensure accurate calibration of the eye-tracking equipment.

Following the instructions, the eye-tracking equipment was calibrated to the participant's position. To assess the referees' gaze behavior, participants were positioned approximately 2 m away from the overhead projector screen. Video clips displayed on the screen were presented without sound to minimize the impact of crowd noise (*Hancock & Ste-Marie, 2013*). Each clip was shown only once during the task.

We selected 31 handball scenes from recorded video footage of matched sanctioned by the Polish Handball Federation during the 2020/2021 season. These scenes, captured using a GoPro Hero 9 camera (GoPro, Inc., San Mateo, CA, USA), depicted on-court actions from the perspective of a goal line referee. Various match scenarios were analyzed, including "Play on" actions, "Free throw", "7-meter", "Punishment" and "Offensive foul". The video clips were chosen based on optimal image quality and relevance to game situations. Footage featuring referees or their ongoing evaluations of the situation was excluded to minimize potential bias in participants' decision-making process. To ensure anonymity, the clips were processed and edited using iMovie software (Apple Inc., Cupertino, CA, USA), concealing the referees and their decisions. Participants were unaware of the selected scenes beforehand.

Each scene was 5 s long and was presented to participants only once. After each clip, the screen went black for 5 s, during which participants provided their responses. The researcher recorded these responses, categorizing them as "Play on/Foul (Free throw)", "Punishment" (yellow card, 2-minute suspension, red card), "7-meter", or "Offensive foul". Participants were instructed to make their final decisions in accordance with official handball rules. An independent panel, consisting of two international referees and one international delegate from the Polish Handball Federation, evaluated the accuracy of the participants' decisions.

## Measures

The video scenes were processed using iMotion 9.1 software (SmartEye Group, Gothenburg, Sweden), which was utilized to analyze data collected by eye-tracker. Participants wore a head-mounted mobile eye-tracker while watching video clips, and their gaze behavior was analyzed. Eye-tracking technology was employed to measure visual parameters such as fixations and saccades of Polish handball referees. The study utilized the Pupil Labs Invisible eye-tracking device, which consisted of two main components: a lightweight head unit (46.9 g) and a recording unit (162 g), connected by a thin cable. The head unit featured protective lenses and a front-facing scene camera with a resolution of $1,088\times 1,080$ pixels. The camera operated at a recording rate of 30 Hz and provided a recording angle of 82° horizontally and 82° vertically. This configuration allowed for the precise capture of gaze behavior while maintaining minimal interference with the participant's natural movements. Additionally, the head unit had two cameras for each eye, which were used to record eye movements at a sampling rate of 200 Hz. Fixations and saccades were extracted automatically using the analysis software provided with the eye-tracking system. A fixation was defined as a period during which the eyes remained stationary within a 1° movement tolerance for a duration equal to or greater than 120 ms. Saccades were identified as rapid eye movements exceeding a velocity of 50 degrees per second. These thresholds were selected based on established standards in eye-tracking research to ensure the reliability and comparability of the results (*e.g.*, *Catteeuw et al., 2009*; *Holmqvist et al., 2011*).

## Statistical analysis

The data were analyzed using Microsoft Office 365 Excel and the data analysis software system Statistica 13.3 (TIBCO Software Inc., https://www.tibco.com/). The analysis was carefully aligned with the research objectives and questions, adhering to a systematic framework. The statistical analysis comprised the following steps:

1. Descriptive statistics: The primary objective was to identify differences in gaze behavior and decision-making among Polish handball referees, distinguishing between males and females and among referees with varying levels of experience. This was achieved by calculating key statistical parameters, including means and standard deviations.

2. Gaze behavior analysis: To compare variables related to fixations and saccades across different types of video clips (game situations) between genders and referee experience levels, two-way ANOVAs were conducted.

3. Decision-making analysis: To examine group differences, the Shapiro–Wilk test was used to assess the normality of the data, while Levene's test was applied to evaluate variance homogeneity. Depending on the results:

- Student's $t$-test (parametric) was applied when the data met the assumptions of normality and homogeneity of variance.
- Mann–Whitney U-test (non-parametric) was used when the assumptions were not met. These testes were crucial for evaluating differences between male and female referees, as well as between referees at different experience levels. Statistical significance was set *at* $p < 0.05$.

4. Effect size measurement: The magnitude of differences within referee groups was assessed using Cohen's $d$, with thresholds defined as small (0.2), medium (0.5), and large (0.8) (*Cohen, 1988*).

## RESULT

Gender differences in fixations and saccades across different types of video clips/game situations (gender × game situations) are presented in Table 2. No significant differences were found between male and female referees in fixation counts, $F_{(4,1561)} = 0.591$, $p = 0.667$, average fixation duration, $F_{(4,1561)} = 0.473$, $p = 0.775$, saccade count, $F_{(4,1561)} = 0.380$, $p = 0.823$, or average saccade duration, $F_{(4,1561)} = 0.413$, $p = 0.799$.

Table 3 presents the results of differences in fixations and saccades between higher- and lower-level handball referees across different types of video clips/game situations (level of referees × game situations). No significant differences were found between higher- and lower-level referees in fixation counts, $F_{(4,1561)} = 0.701$, $p = 0.590$, average fixation duration, $F_{(4,1561)} = 0.227$, $p = 0.923$, saccade count, $F_{(4,1561)} = 0.210$, $p = 0.932$, or average saccade duration, $F_{(4,1561)} = 0.298$, $p = 0.879$.

The results of decision-making accuracy among male and female referees, as well as between higher- and lower-level handball referees, are presented in Table 4. Male referees demonstrated higher decision-making accuracy in the "Free throw" category $(M = 4.45, SD = 1.66)$ compared to their female counterparts $(M = 3.45, SD = 1.13, p < 0.05$; Cohen's $d = 0.638)$.

Higher-level referees exhibited significantly greater decision-making accuracy compared to lower-level referees. On average, higher-level referees made a significantly higher number of correct decisions $(M = 18.03, SD = 3.68)$ compared to lower-level referees $(M = 15.75, SD = 2.79, p < 0.05$; Cohen's $d = 0.678)$. Notably, the most substantial difference in decision-making accuracy between referee levels occurred in game situations related to punishment scenarios. Higher-level referees demonstrated a significantly greater number of correct decisions in these scenarios $(M = 5.81, SD = 1.19)$ compared to lower-level referees $(M = 4.20, SD = 1.40, p < 0.001$; Cohen's $d = 1.407)$.

## DISCUSSION

The aim of this study was to investigate differences in the gaze behavior and decision-making processes of male and female handball referees officiating at various levels. We hypothesized

**Table 2  Differences in gaze behavior between male and female handball referees concerning different game situations.**

| Game situations | Women M ± SD | Men M ± SD | Interaction effect (gender × game situations) P value |
|---|---|---|---|
| **Fixations (count)** | | | |
| Play on | 20.52 ± 2.40 | 20.82 ± 4.23 | |
| Offensive foul | 18.89 ± 2.21 | 19.99 ± 3.80 | |
| Free throw | 19.83 ± 2.17 | 19.95 ± 3.12 | 0.667 |
| 7-meter | 19.27 ± 2.21 | 19.89 ± 3.11 | |
| Punishment | 20.81 ± 1.63 | 20.21 ± 3.68 | |
| **Average fixations duration (ms)** | | | |
| Play on | 171.76 ± 50.57 | 202.92 ± 51.55 | |
| Offensive foul | 194.13 ± 57.37 | 208.71 ± 68.76 | |
| Free throw | 192.45 ± 52.54 | 210.44 ± 58.17 | 0.755 |
| 7-meter | 189.72 ± 49.30 | 203.31 ± 48.68 | |
| Punishment | 169.47 ± 46.20 | 196.98 ± 51.15 | |
| **Saccades (count)** | | | |
| Play on | 50.08 ± 14.82 | 40.77 ± 12.87 | |
| Offensive foul | 44.45 ± 16.41 | 38.67 ± 10.68 | |
| Free throw | 44.69 ± 15.64 | 38.20 ± 9.89 | 0.823 |
| 7-meter | 44.27 ± 12.52 | 38.21 ± 9.28 | |
| Punishment | 50.08 ± 16.03 | 42.16 ± 11.18 | |
| **Average saccades duration (ms)** | | | |
| Play on | 26.43 ± 4.75 | 27.06 ± 1.48 | |
| Offensive foul | 27.15 ± 6.04 | 27.04 ± 1.53 | |
| Free throw | 27.48 ± 6.29 | 27.25 ± 1.61 | 0.799 |
| 7-meter | 26.25 ± 4.25 | 27.29 ± 1.59 | |
| Punishment | 26.35 ± 4.13 | 28.15 ± 1.64 | |

**Notes.**

M, mean; SD, standard deviation.

that there would be no significant disparities between male and female referees in terms of gaze behavior and decision-making accuracy. This hypothesis was supported, as no significant differences were found in gaze behavior or overall decision-making accuracy. These results suggest that gender does not systematically influence refereeing performance in handball. While previous studies have reported potential gender-based differences in visual search strategies, such as more exploratory patterns in females (*Sammaknejad et al., 2017*), our results indicate that such differences are not evident in the context of handball officiating.

Saccade and fixation patterns are processes unconsciously by the human brain (*Glimcher, 2003*). However, our findings revealed no significant gender-based differences in these aspects of gaze behavior, aligning with our hypothesis. These findings contrast with prior studies on gender differences in eye movements (*Abdi Sargezeh, Tavakoli & Daliri, 2019*; *Sammaknejad et al., 2017*). In those studies, the authors reported that females engage in a

**Table 3  Differences in gaze behavior between higher and lower-level handball referees concerning different game situations.**

| Game situations | Higher-level referees M ± SD | Lower-level referees M ± SD | Interaction effect (referee level × game situations) P value |
|---|---|---|---|
| Fixations (count) | | | |
| Play on | 21.30 ± 4.30 | 19.90 ± 3.07 | |
| Offensive foul | 19.66 ± 3.53 | 19.89 ± 3.62 | |
| Free throw | 20.17 ± 3.06 | 19.53 ± 2.74 | 0.590 |
| 7-meter | 19.88 ± 3.16 | 19.55 ± 2.62 | |
| Punishment | 20.60 ± 3.45 | 19.93 ± 3.22 | |
| Average fixations duration (ms) | | | |
| Play on | 193.71 ± 50.22 | 200.05 ± 56.88 | |
| Offensive foul | 202.99 ± 68.32 | 209.56 ± 64.30 | |
| Free throw | 205.67 ± 62.66 | 207.94 ± 48.40 | 0.923 |
| 7-meter | 199.73 ± 51.57 | 201.39 ± 45.03 | |
| Punishment | 186.36 ± 49.31 | 198.31 ± 53.90 | |
| Saccades (count) | | | |
| Play on | 43.82 ± 13.99 | 41.16 ± 13.62 | |
| Offensive foul | 40.50 ± 12.99 | 39.01 ± 11.11 | |
| Free throw | 40.17 ± 11.72 | 38.72 ± 11.41 | 0.932 |
| 7-meter | 40.29 ± 10.38 | 38.31 ± 10.17 | |
| Punishment | 45.33 ± 12.47 | 41.60 ± 12.87 | |
| Average saccades duration (ms) | | | |
| Play on | 26.72 ± 9.04 | 27.25 ± 7.85 | |
| Offensive foul | 27.49 ± 9.88 | 26.41 ± 7.54 | |
| Free throw | 27.44 ± 10.40 | 27.09 ± 7.95 | 0.879 |
| 7-meter | 26.79 ± 9.18 | 27.50 ± 9.24 | |
| Punishment | 27.38 ± 9.00 | 28.36 ± 10.14 | |

**Notes.**
M, mean; SD, standard deviation.

more extensive and directed search compared to males. Moreover, males and females were found to differ significantly in saccade counts (*Abdi Sargezeh, Tavakoli & Daliri, 2019*).

Focusing on decision accuracy, female referees demonstrated notable differences in free throw game situations compared to their male counterparts. This finding is particularly noteworthy, as referees must decide whether to stop the game with a free throw or allow play to continue. In our study, differences were observed only in the decision-making processes of referees during these scenarios.

Additionally, *Souchon et al. (2004)* investigated penalties called by male referees on both male and female handball players, revealing that female players were more likely to face sanctions than their male counterparts. In contrast, our study focused on the decision-making processes of both female and male handball referees. Our findings revealed no significant differences between male and female referees in terms of gaze behavior or decision-making accuracy, supporting our hypothesis. These results align with

**Table 4 Differences in decision-making accuracy between male and female and between higher and lower-level handball referees concerning different game situations.**

| Game situations | Women | Men | *P* value | Higher-level referees | Lower-level referees | *P* value |
|---|---|---|---|---|---|---|
| | M ± SD | M ± SD | | M ± SD | M ± SD | |
| | | | Correct decisions (numbers) | | | |
| Average/Total | 16.64 ± 2.42 | 17.28 ± 3.78 | 0.598[a] | 18.03 ± 3.68 | 15.75 ± 2.79 | 0.022[a] |
| Play on | 3.36 ± 1.36 | 2.73 ± 1.40 | 0.184[b] | 2.84 ± 1.42 | 2.90 ± 1.41 | 0.741[b] |
| Offensive foul | 2.55 ± 0.93 | 2.43 ± 1.03 | 0.876[b] | 2.48 ± 1.03 | 2.40 ± 0.99 | 0.656[b] |
| Free throw | 3.45 ± 1.13 | 4.45 ± 1.66 | 0.047[b] | 4.48 ± 1.63 | 3.85 ± 1.53 | 0.171[b] |
| 7-meter | 2.09 ± 0.83 | 2.50 ± 0.78 | 0.137[b] | 2.42 ± 0.89 | 2.40 ± 0.68 | 0.858[b] |
| Punishment | 5.18 ± 1.33 | 5.18 ± 1.55 | 0.876[b] | 5.81 ± 1.19 | 4.20 ± 1.40 | <0.001[b] |

Notes.

M, mean; SD, standard deviation.

[a]Student's *t*-test.

[b]U Mann–Whitney test.

prior research suggesting that gender does not systematically influence performance in refereeing roles. This underscores the importance of focusing on individual expertise and cognitive processing rather than gender-based assumptions in the analysis of refereeing performance.

Interestingly, our study revealed that female referees demonstrated a preference for fewer game interruptions compared to their male counterparts. However, this preference did not consistently align with correct decision-making. *Souchon et al. (2004)* and *Souchon, Livingstone & Maio (2013)* also demonstrated that male referees tend to apply sporting sanctions, such as awarding free kicks or free throws, more frequently to female players than to male players, irrespective of the competition level (*Souchon et al., 2004*; *Coulomb-Cabagno, Rascle & Souchon, 2005*). Upon analyzing these results, one may question whether there is a discernible trend in the decision-making processes of female and male referees when evaluating female and male competitions. However, to the best of our knowledge, this remains a preliminary observation derived from our findings, as we were unable to identify existing studies that explore this topic in depth. In our research, we exclusively utilized video clips from only men's matches, which could have influenced the results we obtained in our study. The number of female referees is significantly lower than that of male referees in the Polish Handball Association. There is also limited literature available on female sports referees. Unfortunately, the unique experiences of female referees in the sporting context are often overlooked. *Hancock et al. (2021)* highlighted this issue, finding that only 1.3% of 386 publications on officials in the last 50 years focused on understanding the experiences of female sport officials. Attributing differences, whether physiological or biological, to gender for non-playing roles such as officiating is unfounded (*Coulomb-Cabagno, Rascle & Souchon, 2005*).

The comparison between higher and lower-level referees in the overall analysis did not reveal any significant differences in gaze behavior. It is plausible that both groups employ similar methods and approaches when analyzing decision-making situations. Referees from both the Super and First Leagues and those from the Second League and

Regional league often face comparable situations on the court. Seminars organized by the Polish Handball Federation contribute to a standardized method of observing these court situations. Furthermore, the quality of decisions made may be influenced by referees' experience and other factors that aid them in effectively managing sporting events.

According to *Haider & Frensch*'s (*1999*) information-reduction hypothesis, this result might suggest that through experience, elite referees have learned to optimize the amount of information they process, neglecting task-redundant cues and selectively focusing on task-relevant information. This implies that visual search might not determine decision-making proficiency. Although somewhat limited, current research suggests that sports officials across different skill levels exhibit similar visual search behavior. This suggests that higher-level referees may possess superior abilities to interpret or categorize visual information, thereby enabling them to make more accurate decisions (*Catteeuw et al., 2009*; *Hancock & Ste-Marie, 2013*; *Schnyder et al., 2014*; *Spitz et al., 2016*).

Our study found a significant difference in decision accuracy between higher- and lower-level referees. This implies that higher-level referees exhibit a heightened capacity to perceive and utilize available visual cues, thereby enhancing the effectiveness of their decision-making processes. This finding aligns with the observations of *Catteeuw et al. (2009)* and *Hancock & Ste-Marie (2013)*, who suggested that gaze behaviors alone may not serve as a definitive discriminator among referees of varying expertise levels. It seems that higher level referees use cognitive processes that allow them to process and extract relevant visual information, perhaps by more effectively comparing what they see to their long-term memory (*Ericsson & Kintsch, 1995*). The quality of past decisions and accumulated experience can influence subsequent choices, potentially leading referees to make faster decisions and/or repeat the same decision. As a result, individuals may find themselves drawn towards familiar decisions, albeit with a more automatic and unconscious process compared to the initial decision-making instance (*Wolfe et al., 2004*).

Referees consider several key factors in decision-making, including perceptual-cognitive skills, communication, and player management (*Cunningham et al., 2014*). Higher-level referees may excel in interpreting or categorizing visual information, enabling them to make more accurate decisions. Greater visual familiarity and recognition of patterns and movements on the court provide referees with a better foundation for accurate judgments. *Loula et al. (2005)* demonstrated that both motor and visual experience positively influence action perception and, more specifically, the visual analysis of other people's actions. Their findings revealed that participants' motor and visual experience with certain movements enhanced their ability to make accurate judgments. These results support the "perceptual experience hypothesis", which posits that increased visual familiarity improves perceptual recognition and enhances sensitivity to kinematic information (*Jackson, Warren & Abernethy, 2006*; *Cañal-Bruland, van der Kamp & van Kesteren, 2010*).

In contrast, *van Biemen et al. (2023)* found that gaze entropy differed when football referees made correct *versus* incorrect decisions. Gaze entropy, which reflects the degree of structure in gaze patterns, was higher during correct decisions, indicating less structured (more random) gaze behavior. Conversely, lower entropy (more structured gaze patterns) was associated with incorrect decisions. This suggests that referees adapt their gaze patterns

in response to challenging on-field situations while relying on stereotypical, repetitive gaze patterns for decisions perceived as straightforward. This explanation diverges from the core premise of the information-reduction hypothesis proposed by *Haider & Frensch (1999)*, which suggests that with practice, individuals learn to discriminate between task-relevant and redundant information, focusing cognitive processing on the most pertinent components. Acording to *Gegenfurtner, Lehtinen & Säljö (2011)*, eye-movements are expected to show longer durations and more fixations/dwells on relevant areas and shorter durations and less fixations/dwells on distractor areas, resulting in high selective attention allocation. Distractors containing bottom-up signals that pull attention in the wrong direction can be largely ignored if the observer is guiding attention in a top-down fashion, based on the observer's goals (*Bichot, Rossi & Desimone, 2005*; *Schoenfeld et al., 2007*; *Martinez-Trujillo, 2011*). In the case of referees, there might be a heightened focus on incorrect decisions, necessitating a closer examination of situations they are uncertain about.

However, in a study by *Drumond Moreira et al. (2020)*, there was no significant difference in gaze entropy or the number of gaze fixations between national and state groups of football referees, though national referees presented shorter gaze fixations than their counterparts. The authors utilized 35-second clips, and the results showed no significant difference in the gaze behavior of football referees. Although longer scene durations allow referees to grasp the context of the situation and make more accurate decisions, it has been argued that expertise differences in gaze behavior are more likely to surface during more realistic tasks (*Dicks, Davids & Button, 2010*; *Kredel et al., 2017*). These differences may arise from variations in study methodologies, particularly between natural sport environments and laboratory-based research.

Key variables include the number and duration of fixations on pertinent areas of interest, the systematicity of scan patterns, and measures of visual span such as latency to the first fixation on relevant areas of interest and saccade amplitude. These factors provide valuable insights into search theories (*Gegenfurtner, Lehtinen & Säljö, 2011*).

The number of video clips used in this study aligns with previous research on perceptual-cognitive expertise differences between professional and amateur football referees (*Hancock & Ste-Marie, 2013*). Although efforts were made to design a more realistic, referee-specific decision-making task using complex and dynamic clips projected onto a large screen, the study was conducted in a controlled laboratory environment. This limitation may impact the ecological validity of the task, as gaze behaviors can differ between laboratory and real-world settings due to varying task constraints (*Dicks, Davids & Button, 2010*).

Recent research by *Schrödter, Noël & Klatt (2024)* highlights the role of game management strategies in officiating, suggesting that elite referees adapt their decision-making approach dynamically based on game context, further reinforcing the importance of cognitive processes beyond gaze behavior. *Schrödter & Klatt (2024)* further examined game context and its influence on referee decision-making, suggesting that officials adjust their decision strategies in response to aggressive game environments, potentially impacting gaze behavior and judgment accuracy. This perspective could explain why experienced

referees in our study exhibited greater decision accuracy without significant differences in gaze metrics.

Furthermore, *Meyer et al. (2022)* investigated gaze strategies in basketball defense and found that experts adjust their fixation patterns based on opponent movements, suggesting that anticipation plays a critical role in decision-making. This aligns with our findings that elite referees may utilize cognitive heuristics and pattern recognition, enabling them to make rapid and accurate decisions without necessarily exhibiting distinct gaze behavior differences.

While video footage is a widely used methodology to address challenges encountered in field studies, it has limitations. Some researchers have questioned the reliability of findings derived from such methodologies (*Renshaw et al., 2019*). *Schrödter et al. (2024)* discussed the limitations of video-based decision-making studies, highlighting that real-world factors such as game flow and physical presence may significantly alter referees' cognitive and visual processing strategies. Similarly, *Helsen, MacMahon & Spitz (2019)* noted that the absence of gaze behavior differences in laboratory settings does not necessarily generalize to field conditions. Our research design minimized distractions, such as time pressure, to enable a focused analysis of gaze behavior. However, this reduction of real-world complexity may limit the representativeness of the results.

Another limitation is the relatively small sample size, particularly the underrepresentation of female referees. While our sample size is larger than that of previous studies, this reflects an inherent characteristic of the profession, as female referees remain significantly underrepresented in team sports officiating. Nevertheless, to the best of our knowledge, this study constitutes the first detailed examination of gaze behavior among handball referees, with the exception of a case study by *Fasold et al. (2018)*, which, however, included only three male handball referees and did not focus on gaze behavior in relation to performance.

A further important limitation of the current study is the focus on a small subset of variables where significant differences were observed. The limited scope of these findings necessitates caution in generalizing the results. Future research should aim to replicate these analyses in more complex, real-world settings to determine whether the identified differences persist and to explore potential contributing factors.

## CONCLUSIONS

The results indicated that higher-level referees exhibited greater decision-making accuracy compared to their lower-level counterparts, while no significant differences in gaze behavior or decision-making accuracy were found between male and female referees. These findings support the hypothesis that refereeing performance is primarily influenced by individual expertise and cognitive processing rather than gender.

Further research should focus on natural game settings to account for contextual factors such as time pressure, noise, and physical exertion, which may influence gaze behavior and decision-making. Moreover, analyzing gaze location during live match scenarios could provide deeper insights into the relationship between gaze behavior and decision-making. Additionally, it seems that other parameters of gaze behavior (*e.g.*, fixation location, dwell time) should be examined in context of decision-making of handball referees.

## ACKNOWLEDGEMENTS

We extend our gratitude to all the referees who volunteered to participate in this study. We also want to thank the Polish Handball Federation who helped with the recruitment of the participants.

### Funding

This work was supported by Young Researchers Development Programme of the Poznan University of Physical Education. The funders had no role in study design, data collection and analysis, decision to publish, or preparation of the manuscript.

### Grant Disclosures

The following grant information was disclosed by the authors:
Young Researchers Development Programme of the Poznan University of Physical Education.

### Competing Interests

The authors declare there are no competing interests.

### Author Contributions

- Jacek Świdwa conceived and designed the experiments, performed the experiments, analyzed the data, prepared figures and/or tables, authored or reviewed drafts of the article, and approved the final draft.
- Stefanie Klatt analyzed the data, authored or reviewed drafts of the article, and approved the final draft.
- Adam Kantanista conceived and designed the experiments, analyzed the data, authored or reviewed drafts of the article, and approved the final draft.

### Human Ethics

The following information was supplied relating to ethical approvals (i.e., approving body and any reference numbers):

Ethics Committee at the Poznan University of Medical Sciences (KB/357/21).

### Data Availability

The data is available at Zenodo: Świdwa, J. (2024). Gaze behavior and decision-making processes of handball referees: Insights from gender differences and expertise levels [Data set]. Zenodo. https://doi.org/10.5281/zenodo.12566091.

### Supplemental Information

Supplemental information for this article can be found online at http://dx.doi.org/10.7717/peerj.19401#supplemental-information.

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
