# Peer review of "Gaze behavior and decision-making among handball referees: exploring gender and expertise differences"

_PeerJ, doi:10.7717/peerj.19401_

## Round 0.1 · original submission · Major Revisions

All three reviewers have a number of concerns to be addressed, and this will require substantial revision. Of particular significance in these comments are questions around the evidence (or interpretation of the evidence) upon which the authors' hypothesis is based, and questions about the statistical analyses (eg. accounting for multiple comparisons).

Reviewer 1 ·

Basic reporting

The authors use clear, professional English. Their literature review is sufficient and instructive. The structure of the paper is fine.

The hypothesis are included but not well explained.

Experimental design

The experimental setup is fine and the authors did a good job setting up the experiment.

Validity of the findings

The empirical approach is not sufficient. The results - as of now - are not robust.

Additional comments

I have extensive comments in the attached document.

Annotated reviews are not available for download in order to protect the identity of reviewers who chose to remain anonymous.

Reviewer 2 ·

Basic reporting

In general, this article presents an interesting topic of the relationship between gaze behavior and decision making in referees. My comments and suggestions (that I hope will help improve the article) are listed below.
Introduction
The introduction is generally well-written. I have several comments:
Line 66-67 – “…these insights are transferable to other team sports…” I think this statement is too general. Not everything is transferable from soccer to handball. Duration of play, size of playing field, number of referees in a game, are all different and therefore, it is likely that many aspects of refereeing in soccer do not transfer to handball. I would revise this statement a bit.
Line 74-75 – unique in what way? I’m not sure I follow your reasoning here.
Line 77-87 – Please emphasize here that Fasold et al., 2018 did not measure performance so the reader will clearly understand the difference between your study and that of Fasold (i.e., measuring decision-making).
Line 92-93 – “…referees in football…” You previously referred to it as soccer. Please be consistent – choose one.
Lines 93-97 – when discussing the offside line in soccer, it is about the assistant referees, and not the primary referees.
Lines 111-114 – Starting with “There was significant relation…” Is this part of the previous study – if not, you might be missing a reference for the study described in lines 109-111.
In addition, please try to revise these sentences as they are difficult to follow.
Lines 122-131 – what is the benefit of discussing softball and gymnastics? How can it relate to handball? I’m not sure this paragraph is needed.
Lines 132-139 – here as well, how does this add to the introduction?
Lines 140-148 – what is the rationale to expect differences in perceptual-cognitive skills of referees between males and females. In previous studies, as you mentioned, males and females were examined together because, I suspect, there is no real reason the expect differences. So why did you decide to look for such differences? If there are differences, please provide a few examples / references (you discussed it a bit in the discussion).
Lines 149-158 – Here you have two claims – first, you want to improve the sample size. That is absolutely fine. But that is not your main claim – throughout the introduction you provide some lit. review and then explain that there are no studies on gaze behavior and decision-making in handball referees – that is your main rationale. Reaching this paragraph and reading the sample size claim seems to appear out of nowhere – I would revise if possible.
In addition, I agree with your hypothesis that there shouldn’t be differences between males and females – which brings me back to my previous comment – what is the rationale to expect such a difference and test it?
If possible, I ask that you present your reasoning better throughout the introduction and leading to the hypotheses.
Finally, your final hypothesis needs justification as well: “higher-level referees would demonstrate longer fixation durations, fewer saccades, and superior decision-making accuracy compared to their lower-level counterparts”. In the introduction you cite studies that found differences in gaze behavior between referees of different skill levels and studies that have not. So why did you choose to hypothesize that there will be differences? You could have taken the other side just as easily couldn’t you?

Experimental design

In general, the experimental design is OK.
Line 191 – were they asked to maintain stable head and neck only for calibration or throughout the experiment. This is important as wearable eye trackers are head-centered and not world-centered. Thus, head movements can affect the reliability of automatic fixation and saccade measures supplied by the software.
Lines 224-232 – how were saccades and fixations extracted. Was it done by manual analysis or automatically by the software – what were the values used to decide on fixations? Duration and field of view (e.g., fixation was above 100 msec and below 1 degree of visual field). You provide values but as a range for duration – what were the detection thresholds?
Lines 244-253 – The assumption required for parametric statistics is that of the normality of residuals – not of the data itself. Did you test for that?

Validity of the findings

You conducted many statistical tests, but you did not mention controlling for multiple comparisons. This should have been done using a process like the False Discovery Rate (Benjamini, Y., & Hochberg, Y. (1995). Controlling the false discovery rate: A practical and powerful approach to multiple testing. Journal of the Royal Statistical Society: Series B (Methodological), 57(1), 289-300.)
If you would do that, the difference in “play on” average fixation duration between men and women would disappear. (line 263). Same for the difference in accuracy (lines 266-268) – after correcting for multiple comparisons it would probably disappear.
Line 279-285 – I believe that the findings of decision accuracy would remain after correcting for multiple comparisons.

Additional comments

Throughout the manuscript and the tables, the number of digits after the decimal point changes.
Sometimes you use two digits, sometimes three, and sometimes more. I would try to be consistent (2 digits are usually enough).
Throughout the manuscript, you sometimes used male/female, and sometimes men/women. Please be consistent.
Discussion:
Lines 293-298 – as mentioned before, after controlling for multiple comparisons there will be no differences between males and females as expected. Therefore, this hypothesis is supported. Discussing the “trends” is irrelevant in my opinion.
Lines 300-313 – I think that you are building a story here on a spurious finding (that, again, would disappear after controlling for multiple comparisons). In addition, if there is such a difference between males and females – why would it appear specifically in the “Free throws”?
Lines 314-332 – I am not sure how this paragraph adds to understanding the results of your study. I would consider deleting it or shortening it considerably.
Indeed, the discussion surrounding males and females should be shortened and changed based on the lack of differences between groups. Then, I would concentrate on the differences between high level and low-level referees.
Lines 388-397 – How does increased entropy fits with the information-reduction hypothesis? More entropy suggests random scanning that could include irrelevant areas in the visual field. So the finding by Van Biemaen et al., 2022, seem to contrast the information-reduction hypothesis. Perhaps I missed something here, could you please explain this better?
Lines 409-423 – I would delete this paragraph as judging gymnastics is very different than refereeing in handball.
Line 447-448 – except for Fasold, 2018.
Please rewrite your conclusion after addressing my comments in regards to the male – female lack of differences.

Reviewer 3 ·

Basic reporting

The study titled “Gaze behavior and decision-making processes of handball referees: insights from gender differences and expertise levels” is an original study that contributes to the scientific literature. The findings and conclusions of the study can contribute to handball referees and federations. The study's strengths are the high number of experienced referees included in the study; the standardized and comparable video-mediated referee decision-making method was used. Weaknesses of the study: The standardization criteria of the videos shown to the referees in the study were not explained. How the referees' decisions during the study were differentiated between correct and incorrect and how the data were evaluated were not clearly stated.
The article was written in English and used clear, unambiguous, technically correct text. The article was expressed in professional standards of courtesy.
The description of the work is sufficient for a clear understanding of the hypothesis, but shortening would be a good idea.

Experimental design

The research has well defined the research question and established a good methodology for it. However, some information that may affect the findings were not mentioned in the metadology of the study. These are;
The criteria for the inclusion of the 31 handball position videos used in the study in the evaluation were not clearly stated,
The correctness or incorrectness of the decisions made by the referees should be evaluated by whom and according to which criteria.
It should be stated whether the video was re-watched in case of a wrong decision (What was done in case of a wrong decision?).
How long it took to make 31 different decisions (equal time for everyone?)
It should be explained whether these decisions were influenced by the referees' fatigue etc. processes.

Validity of the findings

The findings are not incomplete and inaccurate as can be seen from the row data. However, the fatigue factor in the decision-making process of handball referees in repeated measurements was not included in the hypothesis test. Is this a situation that may affect the results of the study?
Can referees show signs of fatigue after 31 decisions?
Does the first decision making time differ from the last decision making time?
In this respect, can the main effect time, timeXrepeat differentiate the result? Can this mode cause a difference in the hypothesis judgement?

---

## Round 0.2 · Minor Revisions

Thank you for your previous responses to the reviewers. Reviewer #1 has asked for additional information. In particular, questions about the statistical analyses remain. It appears that you may have performed these analyses and not included them in the manuscript? Whether you have performed these analyses or not, they should be included in the paper as described by the reviewer. Questions remain regarding how the referees were classified, and who was responsible for the classification. To what extent are there overlaps in the league assignments, and does that impact on the classification scheme used in the paper?

Reviewer 1 ·

Basic reporting

All good. Nothing to complain.

The authors have hypotheses that fit. However, as they did not preregister their study it is straightforward to find hypotheses afterwards that fit to the study.

Experimental design

See comments in section 4.

Validity of the findings

See comments in section 4.

Additional comments

Dear authors,

Thank you for your revision. I understand that you put a lot of time and effort into revising your paper. I appreciate the changes you have made. Below are my comments.

1. You write that you did a two-way ANOVA. I would like to see this model. Could you please include this in your rebuttal letter and explain why this model is worse for your analysis than model 3 and model 4.

I would have chosen a regression with a binary dependent variable (also probit or logit), but I am interested to see what your ANOVA model shows - please include a detailed description.

2. In the text, you include a nice description of your variables. For example, lines 321-322 “average saccade duration, F(4,1561) = 0.413, p = 0.799”. It would be easier for the reader if you included some more information (such as the mean, the difference, the name of the test you first mentioned, and the p-value). For example, (20.8 vs. 19.4; Average Treatment Effect (ATE) 1.4; T-test, t = -1.23, P = 0.00) - these numbers are just an example. This would make it easier for the reader to understand your data without looking at the data.
3. You write in your rebuttal letter “In Poland, the Super League and First League were classified as more professional leagues compared to the Second League and Regional level of competition.” Please clarify who classified the leagues in this way and provide a reference. Does this mean that all referees in the First League also referee in the Super League? And also that all Second Division referees officiate in Regional Leagues?

Reviewer 2 ·

Basic reporting

The authors have revised the manuscript to my satisfaction. I have no further comments.

Experimental design

n/a

Validity of the findings

n/a

Additional comments

n/a

Reviewer 3 ·

Basic reporting

Academic English is used in the article. The findings are rigorously compared with previous research and the results are related to theoretical models (e.g. the information reduction hypothesis). The raw data is shared and the tables are professional. The number of participants (51 reviewers) was larger than in previous studies. The use of eye-tracking devices was supported by objective measures such as gaze duration and saccade analysis.

Experimental design

The study makes original contributions to the study of refereeing in sport by examining the gaze behaviour and decision-making processes of handball referees. This is an understudied area in the literature. Experiments conducted in an ideo-based laboratory environment provided controlled conditions, allowing detailed analysis of eye movements. This made it easier to reach more robust conclusions. As a recommendation, the study was conducted only with referees belonging to the Polish Federation of Referees. The influence of cultural or structural differences may have been underestimated.

Validity of the findings

The study was analysed using powerful statistical tools such as two-way ANOVA, t-tests and Cohen's d. This increases the reliability of the results. As a recommendation, the number of female reviewers (n=11) is low, which may limit the results on gender differences.

Additional comments

This study meets high standards, especially in terms of methodology and data analysis. The finding that refereeing decision processes are linked not only to visual attention but also to experience makes a meaningful contribution to the literature. However, the study could be enriched with larger samples, observation in the field environment and analysis of environmental factors such as stress. These additional studies would allow for a more holistic understanding of umpires' decision-making mechanisms.
When completed, this research has the potential to be one of the reference studies not only in handball but also in other fast-paced team sports.

---

## Round 0.3 · accepted · Accept

Thank you for your thoughtful responses to the reviewers.